# Lived Experiences and Perceptions of Childbirth among Pastoralist Women in North-Eastern Ethiopia: A Multimethod Qualitative Analysis to the WHO Health Systems Responsiveness Framework

**DOI:** 10.3390/ijerph182312518

**Published:** 2021-11-28

**Authors:** Nejimu Biza Zepro, Araya Abrha Medhanyie, Afework Mulugeta Bezabih, Natalie Tarr, Sonja Merten

**Affiliations:** 1College of Medicine and Health Sciences, Samara University, Samara 7260, Ethiopia; 2Department of Epidemiology and Public Health, Swiss Tropical and Public Health Institute, Kreuzstrasse 2, 4123 Allschwil, Switzerland; natalie.tarr@unibas.ch (N.T.); sonja.merten@swisstph.ch (S.M.); 3Medical Faculty, University of Basel, Klingelbergstrasse 61, 4056 Basel, Switzerland; 4School of Public Health, College of Health Sciences, Mekelle University, Mekelle 1871, Ethiopia; araya.medhanyie@gmail.com (A.A.M.); afework.mulugeta@gmail.com (A.M.B.); 5Center for African Studies, University of Basel, Rheinsprung 21, 4056 Basel, Switzerland

**Keywords:** skilled birth attendance, sexual and reproductive health, health system responsiveness, pastoralism, phenomenology, Afar, Ethiopia

## Abstract

Maternity should be a time of hope and joy. However, for women in pastoralist communities in Ethiopia, the reality of motherhood is often grim. This problem is creating striking disparities of skilled birth uptake among the agrarian and pastoral communities in Ethiopia. So far, the depth and effects of the problem are not well understood. This study is intended to fill this research gap by exploring mothers’ lived experiences and perceptions during skilled birthing care in hard-to-reach communities of Ethiopia. An Interpretive Phenomenological approach was employed to analyse the exploratory data. Four key informant interviews, six in-depth interviews, six focus group discussions, and twelve focused observations were held. WHO responsiveness domains formed the basis for coding and analysis: dignity, autonomy, choice of provider, prompt attention, communication, social support, confidentiality, and quality of basic amenities. The skilled birthing experience of nomadic mothers is permeated by a deep-rooted and hidden perceived neglect, which constitutes serious challenges to the health system. Mothers’ experiences reflect not only the poor skilled delivery uptake, but also how health system practitioners are ignorant of Afar women’s way of life, their living contexts, and their values and beliefs regarding giving birth. Three major themes emerged from data analysis: bad staff attitude, lack of culturally acceptable care, and absence of social support. Nomadic mothers require health systems that are responsive and adaptable to their needs, beliefs, and values. The abuse and disrespect they experience from providers deter nomadic women from seeking skilled birthing care. Women’s right to dignified, respectful, skilled delivery care requires the promotion of woman-centred care in a culturally appropriate manner. Skilled birthing care providers should be cognizant of the WHO responsiveness domains to ensure the provision of culturally sensitive birthing care.

## 1. Introduction

Ethiopia is a multi-ethnic nation, with more than 80 ethnic groups. The estimated population in pastoral areas is 15 million, cover two-thirds (60%) of the landmass of the country. The majority of people the reside in the Somali, Afar, and Oromia regions [1,2,3]. Nevertheless, pastoral communities have received less developmental attention, even though pastoralism makes a vital contribution to the Ethiopian economy. The provision of basic health services adapted to the nomadic way of life is challenging due to several factors: remoteness from the nearest health facility, absence of functional mobile clinics, and lack of information on the part of pastoralists on where to go and whom to consult. The uptake of primary healthcare services like skilled childbirth among nomadic pastoralists in Sub-Sahara Africa is very limited compared to the general population [3,4,5].

Maternal mortality is most common amongst rural marginalized women from low socioeconomic groups, like nomadic women [6,7]. One of the factors contributing to high maternal mortality is the low uptake of skilled birth attendants (SBAs), which constitutes only 15% in the Afar region [8]. Efforts to increase uptake need to target and bring on board vulnerable populations. Goal 3 of the SDGs focuses on ensuring good health and well-being for all, aiming to drop the global maternal mortality ratio and ensure universal access to sexual and reproductive health services by 2030, while Goal 10’s focus is on reducing inequalities. These global targets face distinctive challenges, influencing women’s access to and uptake of health services, with one of the most prominent being mistreatment by care providers [9,10]. SBA is one of the best means to reduce maternal mortality [11]. However, women using SBA services are also highly subject to unresponsive care determinants such as verbal abuse and other forms of disrespect from caregivers [11,12].

SBAs are health professionals who have the minimum skills and knowledge necessary to assist normal childbirth and are trained to be able to identify, manage, and refer any complications for mother and baby on to other more skilled and/or better equipped professionals. Babies can also be delivered with the assistance of doctors, nurse/midwives, and health officers with life-saving skills. SBAs are educated, trained, and qualified maternal and newborn healthcare providers in accordance with national and international standards [13,14,15].

Globally, women suffer from marginalization due to their gender. However, the case of pastoralist women is more serious. They suffer from the double marginalization of being women and nomads. Thus, women’s perception of health care services and an understanding of their service utilization is key to improve the responsiveness of care of the service providers and increase service uptake by women [16].

The concept of health system responsiveness (HSR) was developed and expressed in the World Health Report 2000 [17,18]. The idea of responsiveness relates to client-reported expertise that focuses on the system’s response to service users’ legitimate expectations [19]. Accordingly, responsiveness has become a serious thought in evaluating the standard of healthcare. It is outlined as meeting the expectations of clients and their families relating to the non-medical aspects of healthcare. It aims to help communities equitably meet core social goals and international human rights. Addressing the legitimate expectations of individuals is a central concern contributing to an improvement of health systems’ services [20,21].

Unwelcoming staff behaviour towards clients was mentioned as a reason for low health-seeking behaviour in general and maternal health services in particular. Some communities do not have a good relationship with the staff of the nearby health facility. They believe maternity staff have a negative attitude, are very rude, and do not attend to them with respect. This scared away mothers and made them deliver their babies at home [16]. Conducting a qualitative study is helpful to investigate people’s experiences and perceptions, as well as to understand what goes on behind the scenes [22,23,24]. Phenomenology is one way to inquire into these lived experiences. This philosophical study assesses the essence of consciousness as experienced from the first-person point of view. This takes the intuitive experience of phenomena that could also be done by imaginatively taking one’s own ‘conscious of’ moments as a basis for trying to understand the context what another’s might be.” This method is an appropriate approach for describing perceptions and experiences as it captures local needs and includes the voices of such explicit populations [24]. 

However, there is a knowledge gap on women’s perspectives of the responsiveness of the care that they receive and perceptions and expectations of service providers. Here, we will explore the views, perceptions, and experiences of birthing mothers with regard to the behaviour and treatment received from skilled service delivery attendants. Thus, we can identify factors that mothers perceive influence the delivery of intrapartum care to the pastoralist communities. In addition, we will explore the extent to which these factors influence their choice of place of delivery. 

## 2. Materials and Methods

### 2.1. Study Setting and Period

The study was carried out in Afar Region, which is located in the north-eastern part of Ethiopia. This region is one of the eleven regional states of Ethiopia. Afaraf is the local language, widely spoken by rural communities. The region has an estimated total population of 1,816,304, of which 799,174 (44%) are females [25]. Pastoralism is the dominant production system in the Afar Region (90%), while agro-pastoralism constitutes the rest (10%) [26]. The region is characterized by an arid and semi-arid climate, with low rainfall. The region is one of the hottest inhabited places in the world, with temperatures exceeding 40 °C and less than 200 mm rainfall per annum. Afar is increasingly drought-prone. A high percentage of the total population is food-insecure and illiteracy rates are high. Health information is rarely available in the Afaraf language. There is low health service coverage, low access to paved roads, and low access to potable water. Maternal health services are poorly equipped and are inaccessible in terms of geographic location, security issues, linguistic barriers, and poor documentation of vital events [27,28,29]. The study was carried out in three districts (Mille, Afambo, and Kori). Women with recent SBAs were selected purposely. The rationale for the sample size in this study focused on obtaining data saturation, where an iterative interview process continued until no new themes emerged and no new insights could be gained with additional data collection. This brought a varied sample, having sufficient textual data to make an iterative categorization of qualitative data possible. 

### 2.2. Study Design 

This research is framed by a larger research project, named Reproductive, Maternal, and Newborn Health Innovation Fund, which is aimed at improving reproductive, maternal, and child health services uptake in pastoralist communities [30]. Qualitative methods are appropriate to capture the voices of specific populations for measurable as well as unmeasurable outcomes. Thus, the phenomena of observed and unobserved experiences, actions, internal meaning, and external consequences can be recorded. It is the best method for describing the perceptions of different populations and their local understandings [31,32]. An Interpretive Phenomenological approach was employed to analyse and interpret the interview texts for the exploratory data [33]. Key informant interviews, in-depth interviews, and focus group discussions were held with purposively selected study populations. 

### 2.3. Study Participants 

A list of registered post-natal women with recent uptake of skilled birth attendance within the past six months was collected from registry report books of the nearby health facilities. A local community coordinator was carefully chosen in consultation with the local health bureau, with the intention that she be capable of convincing women to participate in the study. This coordinator was well known and led the locally established health development army. She was a respected, long-standing member of the community and well aware of the local language and culture. This local coordinator contacted eligible post-natal women to invite them to participate in the study. Under her guidance, a transect drive was conducted at each site to approach mothers with recent experience with skilled delivery service uptake to volunteer to take part in the study. A total of 36 participants volunteered for the study. 

### 2.4. Data Collection Tools and Procedures 

Data collection was done through four key informant interviews, six in-depth interviews, six focus group discussions, and twelve focused observations. The key informants were health experts who were chosen because of direct involvement in the health facility, either as health service providers or as facility administration. The in-depth interviews were done with pastoral women, who were chosen for their knowledge and experience of maternal health services and their dynamics. The focus groups also consisted of pastoral women who were targeted on the basis of a recent experience with skilled delivery service uptake in the nearby health centre. The focused observations were done to observe health providers’ approach to care for labouring women, room hygiene, and cleanliness of basic amenities. This was possible with a structured observation checklist. Each observation took 25 to 40 min, on average. All participants were informed about the study purpose and all of them provided oral consent to participate in the study.

Trained research assistants, who were health professionals, had proven experience in qualitative data collection, and were well aware of the local language and culture, were chosen to moderate the data collection under the strict supervision of the principal investigator and supervisors. All interviews and FGDs were audio-recorded, transcribed in the Amharic language by the research assistants, and translated into English by the principal investigator. We used a semi-structured interview guide to ensure that important discussion topics were not missed. 

The WHO health system responsiveness framework, [16,17,19,21,34,35] was used as a source to develop the interview guide. The knowledge gained from the analysis of each interview was used to modify the consecutive interview guide. Interviews were performed continually until data saturation was achieved to the point that no new code could be extracted. 

A minimal compensation for travel expenses was paid in cash to study participants. Snacks and soft drink refreshments were provided to keep the participation active.

### 2.5. WHO Framework of Health System Responsiveness 

Key findings were mapped against the framework of the wider health system responsiveness domains that are used to promote health service uptake. This WHO framework consists of eight themes for health system responsiveness: dignity, autonomy, choice of provider, prompt attention, communication, social support, confidentiality, and quality of basic amenities [16,34]. Health responsiveness–related factors that stop pastoralist women’s from SBA were explored.

### 2.6. Data Analysis

Data analysis was conducted according to the principles of interpretive phenomenological analysis (IPA), where the researcher attempts to understand the interview and observations from the participants’ (or interviewees’) perspective [33]. The domains of the health responsiveness framework [16] were used to establish coding with emerging and re-emerging themes, with the help of Atlas.ti version 7.0 (Atlas.ti Scientific Software Development, GmbH Berlin, Germany) to organize the data. The codes were able to capture perceived experiences and perceptions towards utilization of SBA services. Key statements of the participants were explored to find recurring patterns, thoughts, feelings, or ideas. The final set of themes was summarized as a codebook; recurring and unique quotes were described. 

In the first stage, we read the translation files several times to capture the women’s experiences, feelings, and perceptions. The total focus for analysis was based on empathy, where the researcher’s previous readings, judgments, and understandings were put to the back of his or her mind (naivety) [33]. Women’s perspectives became the key to understand the depth of the subject (bracketing). In the second stage, we identified important key phrases in the text document. Then, in the third step, we extracted the concepts, and in the fourth, we categorized concepts into groups based on similarity. In the fifth stage, we combined the categories to form concepts that explain the development of classes that are more general in terms of categories. In the sixth stage, we presented a comprehensive description of the structure of the phenomena under study. In the final stage, we validated our interpretive phenomena by comparing them to the standard HSR domains of the WHO.

### 2.7. Quality Control

The research team, consisting of research assistants (data collectors), local coordinators, and supervisors, was extensively trained in qualitative data collection techniques. Supervisors, together with the principal investigator, conducted regular feedback and follow-up sessions. The quality of this study was assured through prolonged engagement of the principal investigator in the fieldwork. We did member checking, where feedback from study participants was taken in the final sessions of each interview before concluding the main points. Two researchers read the files several times and performed coding, clusters, and themes independently. We also included peer debriefing among the research team. We allotted time to the specifics of the interviews and the review of conflicting cases to improve credibility. Qualified peer researchers were invited to review and assess transcripts, emerging and final categories from those transcripts, and the final findings of a given interview. 

Data triangulation was done via the application and combination of several research methods (interview, observation, and literature review). Various disciplines in the research team and previous experience allowed a high level of reflexivity, which challenged us to examine and address assumptions and biases. We examined the similarity of the extracted themes and theme clusters to those extracted by HSR categories. There was almost 80% agreement in coding and theme formulation with those extracted from HSR domains. In the case of a discrepancy, the research team reviewed the file and re-analysed the disagreement. 

### 2.8. Ethical Approval

The study was conducted according to the guidelines of the Declaration of Helsinki; ethical permission and support letters were obtained from Mekelle University School of Public Health (with reference letter CHS/498/SPH/11). Permission was also secured from the Afar regional health bureau and respective district health offices. The purpose and aims of the study and researchers’ expectations of participants were explained to study participants by the research team. The information sheet was read to participants in the local language so they could understand their rights. The issue of confidentiality was addressed by using unique identification numbers. Verbal informed consent was sought from the study participants. The interviews were conducted individually and in private. The FGDs were held in three district health centre meeting rooms. All information collected, including the audio files, was kept confidential in password-protected files with personal computer access; participant identities were coded to keep all data anonymous.

## 3. Results

### 3.1. Socio-Demographic Characteristics of Study Participants 

All focus group discussants (FGDs) were women with recent SBAs. Two age groups (30–39 years and 40–49 years) accounted for all (100%) of the total FGDs. A significant proportion of in-depth interviewees (IDIs) and FGDs had no formal education (Table 1, Table 2 and Table 3).

### 3.2. Experience and Perceptions of Skilled Birthing Care

Experience and care perceptions of skilled birth service users were grouped under three major categories: cultural acceptability of care, responsiveness of care, and preference to skilled birthing modalities (Table 4).

### 3.3. Health System Responsiveness Domains 

#### 3.3.1. Dignified Care

Negative staff attitude was mentioned as a reason for the low use of maternal health services in general and skilled childbirth services in particular. Study participants stressed that the communities did not have a good relationship with service providers. They said maternity care providers showed unfriendly behaviour towards women in labour and did not attend to women with respect and compassion. A few SBAs were reported to be rude, which scared away mothers and made them decide to deliver at home. Quotes from study participants capture the issue of perceived bad attitude: 

  *“There are health workers at our health facility who are rude, at times they shout at mothers, and they are also incompetent to give proper care during labour”*.(FGD 2, para 3)

Discussants complained about negative attitudes from maternity staff. The complaints included shouting at a mother for not pushing well enough during labour. Some maternity care providers simply commanded a woman; they often belittled and shouted at clients, especially women from rural areas. Only a few health professionals provided respectful care, women reported. Pastoralist women coming from remote rural areas felt discriminated against because of their wet clothing, giving off a bad smell. Discrimination experienced by the women from care providers included refusal to provide a physical examination, rude behaviour like shouting, frowning, discriminatory speech, and unnecessary referrals. These negative attitudes of the staff discouraged mothers from using skilled childbirth services. In addition, fear of discrimination on specific personal experiences, especially in relation to female genital cutting (FGC), were expressed by the women, who experienced a large proportion of SBAs as unable or unwilling to perform FGC care or de-infibulation.

A woman narrated her lived experience: *“…while everything was going wrong, relatives took me to the nearby health centre with wasaka (people used to bear vulnerable clients on their shoulders in a conventional way). We lost many hours in the health centre just waiting for a health professional. When he finally came, he just gave us a referral paper to go to Dubti hospital, which is very distant from here.”* (IDI, Para 3). She continued by saying, *“The doctor at the hospital told me that the foetus had already died and that my uterus had ruptured… Families were asked to donate blood to help with the surgery … emmmm… I felt bad. [Long silence] I can’t talk now [Crying]. That’s it! …[wiping eyes with her headscarf]. It is not only losing dignity but also faith. It is a double punishment for me. When I went back home, my mother was the first person to cry. I also witnessed my husband’s furrowed face. So all the benefit [we got] from skilled birth attendance was the emotional disturbance developed inside my mind”* (IDI, para 3).

#### 3.3.2. Autonomy

Women in labour generally expect to be examined and their voices to be heard. However, women are not involved in decision-making processes concerning procedures to be applied during childbirth. They are given little consideration to contribute their ideas/opinions on therapeutic procedures such as blood pressure measurement, per-vaginal examination, or birthing position. Not being examined is seen as a departure from norms and autonomy. A mother of four narrated: *“I went to the hospital with severe abdominal pain for the birth of this child (pointing to her newborn). I was told to go home and wait until true labour starts. They never did a thorough examination either. They were not willing to hear my voice either. Immediately upon arrival back home, I delivered”* (IDI, para 4).

Women diagnosed with false labour need deep counselling. One woman shared her experience:

  *“I have travelled such a long distance to visit the health centre for my labour pains. I waited for long hours to see a health professional for my diagnosis. However, they gave me a prescription paper to buy a painkiller from a nearby drug store. I was very upset and regretted having travelled all the way, wasting too much time at the health centre. Had I known that I would get only prescription notes. Consequently, I decided not to visit the health centre again”*.(FGD 5, para 3)

#### 3.3.3. Choice of Provider

Women in pastoralist communities opt for care by female birth attendants. This issue of choice of skilled childbirth attendant was explained in this way: *“…Women in my locality frequently explain that anger, sadness, and shame accompany a loss of culture and being distant from Allah’s (God) laws about being seen naked by strangers (male midwife) in the health facility. Male birth attendants are also involved in assisting women in labour. That is quite against our faith and culture. God prohibits us from doing so. … I fear this will cause punishment from God; Almighty Lord”* (IDI, para 3).

This choice of service provider is often associated with exposure to per-vaginal examinations. This procedure is viewed as disgraceful, frequent, and painful in the pastoralist community. Women commonly mentioned this to be an inhibiting factor to their use of institutional skilled child delivery services. One FGD discussant mentioned, “*Women do not like health facility deliveries. Per-vaginal examinations are frequent and disgraceful. In my last pregnancy at the health facility, a male birth attendant kept on pushing his fingers into my private parts (vagina)… ehhhh… it was very painful, shameful, and embarrassing. I wonder what they were looking for*” (FGD 3, para3). Another discussant also narrated, *“During my recent delivery, I was exposed to a series of per-vaginal examinations. It was my first delivery in the health facility, and then I decided not to go there again even for child immunization. It is disgraceful to allow a male stranger to see me naked, let alone to insert his fingers into a woman’s private parts”* (FGD 2, para 1).

#### 3.3.4. Prompt Attention

In addition, the long waiting time reported by women who frequented health facilities discouraged them from delivering there. Unreasonable delays in the health facilities were consistently mentioned as a deterrent to using skilled child delivery. A woman narrated her frustration as follows: *“We go to the health facility as early as we feel first sense of labour… We spend half a day waiting to be attended to … the health professionals were not on duty”* (FGD 2, para 3).

Other women experienced severe delays in receiving attention in the delivery room and felt neglected and sometimes exposed to risk: *“**I was in labour. Instead of accompanying me, the nurse was just watching a movie until relatives knocked on her door”* (FGD 6, para 2). Another woman revealed, “*I was told to wait in a health centre for over two hours, despite being in severe pain and receiving no explanation from the maternity staff”* (FGD 7, para 4). Women’s experiences with the health system tended to influence their future use of health care services. Women in particular openly talked and gossiped with neighbours about their experiences with the health care system. These included interaction with care providers, examinations, long waiting times, and the poor quality of services.

#### 3.3.5. Communication

The majority of maternity staff in the region have a problem understanding and communicating in the local language, Afaraf. This absence of smooth, two-way communication resulted in experiences that are more negative for women. One key informant explained, “The majority of health professionals have a problem with understanding the local culture and speaking the language (Afaraf). Almost none of the professionals you see in this facility are originally from the local community (Afar), which makes two-way communication very difficult” (KII, age 35). The relationship between women and maternity staff seemed to be fundamental to the experience of birthing care, with better relationships resulting from experiences that are more positive. Some research participants mentioned that without an open relationship with skilled birth attendants, they felt unable to communicate their concerns and to discuss health issues with them. Even though women felt they needed or wanted to give birth in the health facility, maternity staff did not provide any information about procedures before delivery. A mother of two shared her experience: *“I went for my first delivery and they didn’t explain to me what to do, they just said open your legs, I didn’t even know how to push and in that way, I found out that my baby had drowned (aspirated) … They were not even able to explain this to me in a language that I can understand”* (FGD5, Para2).

#### 3.3.6. Social Support during Labour

Social support and companionship during labour and delivery are highly valued by pastoralist women. However, skilled delivery services do not allow pregnant women to bring a companion of choice into the birthing room. Since there is no restriction on the number of supporting companions during birth at home, women choose to deliver at home, despite the government’s encouragement of delivery with SBAs. The testimony of a pastoralist woman highlights this issue: *“In the health facility, families and relatives are not allowed to accompany the women; they only come when the baby comes out, if you are lucky”* (FGD 4, para 3).

Trained Birth Attendants (TBAs)—*uletina* in Afaraf—are well known for the social support and accompaniment they offer during labour and delivery. In one IDI, a witness expressed it in these words: *“When my term approached to labour, my husband took me to the TBA’s house. I stayed there for two months … she is kind and well experienced … she did not push her fingers into my private parts… she encourages labouring women to push. She also accompanies women to the health facility in case of complications”* (IDI, para 3).

#### 3.3.7. Quality of Basic Amenities 

Study participants complained about the poor quality of basic amenities in the health facilities: a dirty room, lack of water in the bathroom, the absence of essential equipment, and the need to share bed linens. A woman said that there had been improvements in cleanliness over time: “*At the time of labour, there were blood drops on the delivery bed, the surface and bed linen smelled bad. There was no water in the facility. For example, I was bleeding; my mother took me to the bathroom, but could not wash me. This year I did not see this. I think there are changes, maybe”* (FGD 2, Para 3). Another study participant mentioned that *“the toilet and shower rooms were filthy and smelled terrible. There was no running water or electricity in the delivery ward. Since the facilities are far from my home, it was a problem to have proper food and drink. My family was challenged with taking care of my food*” (IDI, para 3).

#### 3.3.8. Confidentiality

Study subjects were not comfortable with some clinical procedures, such as birthing position and per-vaginal examinations. A 39-year-old woman stated in an IDI, *“*…*when my neighbour gave birth in the facility, the midwife invaded her privacy and conducted too many vaginal examinations, which was dehumanizing and shameful. So, how am I able to use the treatment room knowing that my friend had a tough experience there? I preferred to give birth at home with the assistance of a TBA, as she can give me more privacy and control over the situation than the midwife at the health facility*” (IDI, Para 6).

#### 3.3.9. Culturally Acceptable Care Perceptions 

Culturally acceptable care is a very important ingredient in enhancing health-seeking behaviour in the nomadic communities in favour of skilled birthing facilities. Many women did not deliver in health facilities because they believed that mothers were not attended to at the level of their traditional and cultural expectations. A 42-year-old woman of four narrated it in these words: *“Immediately following delivery, the neonate, both a boy or a girl, should swallow the sweet part of the date palm mashed with the saliva of a well-respected brave man in the community. This is the culture in Afar… very helpful for the baby’s safe growth and to be a hero for his clan. But this service is not available in the health facility… because of that reason, women prefer home deliveries”* (FGD 4, Para 4). Some FGD participants pointed out that delivery with SBA is only recommended in case a woman needs special treatment. Thus, women are encouraged to give birth at home with the assistance of TBAs or their mothers-in-law. One 32-year-old participant in an FGD commented, *“*…*birth is a natural life event, not a disease*. *Our foremothers were doing it, what is special now? I don’t see a reason to opt for SBA”* (FGD 2, Para 3). Similarly, another study participant spoke about cultural issues related to the handling of the umbilical cord. Some segments of Afar tradition require that the umbilical cord of a baby be taken home by the mother to be buried at home. Otherwise, the mothers believe, the umbilical cord might be eaten by a dog or a cat, which is evil for the final fate of the newborn. 

#### 3.3.10. Alternative Birthing Modality 

A squatting birthing position is pastoralist women’s preferred birthing position, and they expect this to be possible and taken into account when in labour. Afari women feel more comfortable with a squatting birthing position, which was also reported as being one reason for the preference of home childbirth. The community has a long-time tradition of utilizing TBAs. They have been the only delivery caregivers for decades, and the community continues to be comfortable with their services. TBAs, according to the participants, perform multiple roles: they give advice to young women, sensitize the community on maternal health issues, provide antenatal care, help mothers at the time of delivery, and administer local medicines and herbs as first aid. The health centre is far away. TBAs massage the labouring women, refer complicated cases, and accompany mothers and their newborns to health centres. TBAs are knowledgeable, experienced, accessible, and offer a range of health services. Women prefer them as birthing attendants to the skilled birth providers, as revealed by one key informant: *“**Women prefer TBAs. They have a better understanding of labour and help us in many ways. If a woman is pregnant, they visit her at home, gently touch the abdomen to see if the baby is fine, and provide all necessary information about pregnancy care. TBAs are also able to solve problems that occur during pregnancy. When labour begins, a TBA is requested to assist the woman in giving birth. Sometimes, when there is a problem and the TBA has tried and failed, she refers the woman to the health facility**”* (KII, age 35). Another FGD discussant added, *“Social support from a TBA is astonishing, she tries to put butter on the abdomen and helps us massage the uterus; they serve cultural food (‘ounur’) and mashed date palm for the new-born. Women need effective support throughout childbirth, and procedures should be described to them. Having relatives and a TBA present during birth gives us the freedom to feel safe through continuous support, which is important to feelings of belonging and to avoid feelings of loneliness and fear”* (FGD 5, Para 4).

## 4. Discussion

The overall analysis from the qualitative study provided an in-depth interpretive contextual understanding of pastoralist women’s institutional birthing journey. Three main themes could be identified, with numerous subthemes. The three main themes were ‘the influence of responsiveness of delivery service providers’, ‘cultural acceptability of delivery care’, and ‘the availability of alternative birthing care modalities’.

### 4.1. Responsiveness of Skilled Birthing Providers 

Although key developments have been made in Ethiopia, the situation in which pastoralist women live is still challenging. Sexual reproductive health services are poorly equipped, inaccessible, and not well documented in the pastoral context. Pastoralist communities obtain the fewest health development benefits of all Ethiopian citizens and are virtually excluded from the health sector improvements aimed at agrarian populations. Social and cultural practices still contribute to the exclusion of pastoralist girls from education, subsequently reducing participation in development activities. Similar findings were reported elsewhere [3,27].

The impact of skilled delivery service providers’ attitude was a repeatedly mentioned reason for the low uptake of SBA services. Some service providers simply command a woman; they often despise, belittle, and shout at clients, especially women from rural areas. Only a few care providers treat labouring women respectfully. Thus, fear of humiliation in the facilities inhibited pastoralist women from seeking skilled birthing care. A similar finding was reported in a similar setting of the poor responsiveness of skilled delivery care providers [3]. In Ghana, delivery care providers attributed low facility utilization to physical abuse, verbal abuse, neglect, and discrimination. Other studies conducted in sub-Saharan Africa and Bolivia found poor responsiveness of maternity staff contributed to low utilization of SBAs [35,36].

Unequal power relations between the women and skilled care providers can explain the abusive practices described by the participants in this study. Health providers’ accountability at the health facilities, women’s choice of the sex of health care providers, as well as the language and cultural differences, played an important role. In contrast, providers’ favourable attitude toward women positively influenced women’s decision to opt for skilled childbirth in Ethiopia and Tanzania [27,29]. Similarly, women in Malawi perceived respect, privacy, and confidentiality of care as important aspects of the care modality [37]. 

Previous experiences with skilled care intermingle with health system choice. Here it seems that social-economic and cultural factors are responsible for the low utilization of skilled child delivery services. Past negative experiences of the women in the health facilities, such as long waiting hours, bad staff attitude, and cultural sensitivity of service, were described as factors considerably influencing acceptability and utilization of skilled birthing services uptake in the Afar region. A study conducted in Malawi also conveyed similar findings. There, women reported various negative experiences at different health facilities when they went for delivery. These included not being examined during labour and after delivery, a poor reception at the labour ward, being left alone without assistance, and no midwife being present to assist with the delivery [37].

As confirmed through focused observation, maternity attendants are not adequately prepared or committed to clients; they show low readiness and are poorly motivated to provide skilled delivery services. This is mostly attributed to the lack of a conducive working environment: low/no incentives, no promotions, no housing provided with the job, inadequate infrastructure, and an absence of necessary/needed medical equipment. In such an extreme working environment, where temperatures exceed 40 °C and there is less than 200 mm rainfall per annum, a certain amount of incentive is expected to compensate for these hardship-prone working conditions. It is agreed, however, that all these challenges should not be reflected in client care. The services should be accessible to women, both in terms of good quality offered and time invested. Whatever the claim, women expect to be treated with due respect and compassion.

### 4.2. Culturally Sensitive Care

Cultural, religious, and traditional contextualization of available health care services are essential prerequisites to service uptake for Afar pastoralist communities. Afar people are conservative in their cultural beliefs and their religious faith, and they adhere to generally agreed upon community values. There are numerous cultural ceremonies for women in labour, and most women abide by them. In particular, women facing difficulties and birthing complications act upon these beliefs as a first solution. If this should fail, they resort to the health facilities; in most cases, it is too late at this point to provide scientific support for the women.

The present study shows that women prefer home deliveries so they can practice their cultural and religious beliefs and rituals. The health facilities are only resorted to when complications arise, which is often too late. Health service providers might question local religious beliefs and practices, adhering to their own ideas about what constitutes normal and acceptable health care modalities. A similar study conducted in similar settings in Ethiopia and Uganda demonstrates that adherence to traditional birthing practices and beliefs was mentioned to be one important determining factor influencing the uptake of health care services [3,38]. In addition, women are required to demonstrate strength and endurance during childbirth to their community. In Uganda, it was found that pregnancy was a test of endurance, and those who delivered without signs of fear were respected [38]. 

Maternity waiting homes (MWHs) are temporary shelters located near a hospital or health centre and are available to pregnant women from rural areas to help overcome distance barriers and allow for quick transportation to health centres [39,40,41] . The WHO endorsed MWHs as impactful interventions to reduce maternal morbidity and mortality. Such shelters have existed in various forms for over 100 years in Europe and North America. In the developing world, the use of MWHs was introduced in rural Nigeria as early as the 1950s. Several studies have showed that MWHs have a demonstrated benefit for increased utilization of SBAs through the provision of culturally acceptable care provision [10,42,43] .

MWHs have shown promising results in terms of bridging the distance barrier between service providers and their clients and in providing culturally acceptable care. More importantly, the community owns MWHs. Lasting solutions need to be found, however, to ensure proper furnishing and maintenance of the facility and to make sure sustainable food sources are available. One study conducted in Zambia reported that MWHs were built to provide a safe space in which expectant mothers could spend a few weeks before delivery. However, families still incurred indirect costs, as the health facilities were not able to cover food and related expenses [35]. 

### 4.3. Alternative Delivery Modalities

Afari pastoralist women have unique needs, which are strongly attached to their cultural and geographical context [44] . These nomadic women feel more comfortable giving birth in a squatting position. This is one reason mentioned for their preference of home deliveries. Women are not comfortable with the delivery position in the health facilities, where they are asked to lay down on their backs to push (lithotomy position). This position facilitates the return of the baby in the direction of the chest from the pelvic bone and prolongs the duration of labour. Women dislike this delivery position for two reasons: first, it exposes their private body parts to the birth attendant, who is often a male stranger, and second, birth at the health facility takes longer, as previously mentioned. This finding is similar to study findings from Ghana, where women only deliver in the lithotomy position. A study done in Bangladesh also showed that women who can actively participate in their own birthing experience prefer a squatting or kneeling position [36,42]. 

TBAs in the pastoral communities are considered to be empathetic in their care for a woman giving birth. Respondents have trust in the skills of TBAs, so women in the community prefer to continue using their services. Communities also mention that TBAs should be supported with training and medical equipment, as different aid organizations have been doing thus far. A study conducted in Afar recommends TBAs to establish a strong referral system to prevent problems of home delivery. Demand and interest in TBAs’ work continue to be high within the communities [3,43]. A similar study conducted in rural Nigeria indicated that pregnant women prefer TBAs because they are more affectionate and caring compared to institution-based SBAs [45]. 

Pastoralist women feel ashamed of their private parts being touched by male midwives during labour and delivery. Many Afari women do not deliver in a health facility because they abhor being attended to by a male midwife. A male midwife in the delivery unit significantly decreases the use of institutional delivery services. This finding is supported by a qualitative study in which a woman said that “it is only God and my husband who have the right to see me naked. In Afari culture, it is considered impolite and unethical to reveal reproductive health organs to strangers” [46]. Women’s poor uptake of skilled birth services is thus directly related to their cultural and religious beliefs and their discomfort with male birth attendants. These same findings were noted in a Zambian study [35] and elsewhere in Africa [38,45,47] . Pastoralist women prefer to be attended to by a female midwife. This is explained by women’s religious and/or cultural reasons—they do not want a man to see them naked [48]. Women demand that the regional government train female midwives to improve skilled birth services. Similar study findings were reported for Kenyan and Somali women [48,49,50]. 

## 5. Conclusions and Recommendations

The study participants mentioned a range of lived experiences they had during skilled childbirth at the respective health facilities. We extracted three major themes from the analysis; these were unresponsive care, lack of a culturally contextualized care approach, and the lack of alternative skilled delivery modalities. The lack of a responsive skilled delivery care during childbirth is a public health issue that requires greater efforts to generate evidence of the different forms of rights’ violations experienced by women during skilled childbirth. It should also inform the enforcement of solutions in policy and practice. Both the women as service users and the maternity staff as service providers should be involved in designing and implementing impactful policies, with a special focus on marginalized, vulnerable groups. This study also reveals signs of resistance (unwillingness, less cooperation), and rough behaviour was also found among healthcare providers by some participants who attempted to challenge the abusive treatment, which can lead to violence being normalized [47].

The health systems responsiveness practice among pastoralist communities must be adapted to the needs of nomadic populations. Maternity services need to be made more culturally, religiously, and socially relevant so that the lives of pastoralist mothers and their babies can be protected and saved. Any procedures undertaken during labour should be done with a clear, informed decision and the agreement of the client. Making health facilities conducive and friendly, as well as adapted to the needs of pastoralist communities, can have a significant impact on health-seeking behaviours and thus the utilization of health services. 

Accountability mechanisms within the health system can provide women with channels for registering complaints, thus contributing to a decrease in disrespectful and abusive attitudes from SBAs during births. 

Culturally appropriate childbirth models, including the preferred squatting and birthing positions, should be incorporated in the training of SBAs; facilities catering to other positions should be provided as well. Preferences, feasibility, and consequences of different birthing models should be studied further. 

Moreover, empowering the women in the community to voice their concerns, employing female healthcare workers with a deep understanding of the local culture and religion, and improving working conditions by strengthening accountability measures will contribute to an increase in SBA service uptake. Encouraging women to report abusive practices, with a special focus on vulnerable and marginalized groups, needs to be part of these innovations.

Even though the WHO discourages training TBAs, they are the main human resource for women during childbirth in rural pastoralist communities. TBAs are still very active in attending to home births in the community [51,52]. A constructive partnership between TBAs and SBAs is necessary to improve institutional delivery services. Re-enforcement and involvement of TBAs will play a positive role in improving the responsiveness of clients and should be used as an entry point to promote the uptake of Skilled Birth Attendants. Training and counselling for pregnant women during antenatal care may enhance the harmony and tranquillity of pregnant women in the community and convince them to use SBAs. 

Further research is needed to examine husbands’ opinions in the decision-making process with regard to SBAs and their role in enhancing the responsiveness, peace, and tranquillity of their pregnant wives in such difficult conditions. The findings of the study give comprehensive information to plan interventions to enhance SBA service delivery. The study also provides detailed knowledge to the government and to policy makers, especially with regard to improve skilled birthing services for mobile community.

## 6. Strength and Limitations of the Study

The study is among the few studies conducted in pastoralist communities; the application of various methods is the strength of this study. The limitation of this study is that the assessment of the perception of women towards SBAs was limited to women who delivered at health facilities only, not at home. Thus, it is difficult to compare the perceptions of women who gave birth at home. Taking the socio-economic background and nature of the study design into consideration, the findings of this study can only be generalized to a similar pastoralist population.

## Figures and Tables

**Table 1 ijerph-18-12518-t001:** Socio-demographic characteristics of FGD participants, Afar region, Ethiopia.

S.No	Characteristics	Categories	No. of Participants	Percentage
1	Age	30–39	12	46%
		40–49	14	54%
2	Educational status	Illiterate	20	77%
		Primary	6	23%
3	Marital status	Married	23	89%
		Unmarried	3	11%
4	Number of children	1	4	15%
		2–3	9	35%
		≥4	13	50%
5	Occupation	Pastoralist	26	100%

**Table 2 ijerph-18-12518-t002:** Characteristics of KII participants, Afar region, Ethiopia.

Code	Age	Sex	Edu. Level	Research Role	Experience
01	49	Female	Diploma	Skilled birth provider (KII)	5 years
02	27	Male	Degree	Health centre head (KII)	4 years
03	42	Female	Diploma	Bureau head (KII)	12 years
04	35	Male	Masters	Bureau head (KII)	7 years

**Table 3 ijerph-18-12518-t003:** Characteristics of in-depth interview (IDI) participants, Afar region, Ethiopia.

Code	Age	Sex	Edu. Level	Occupation	Marital Status	No. of Children
01	49	Female	Illiterate	Pastoralist	Married	8
02	37	Female	Illiterate	Pastoralist	Married	4
03	42	Female	Illiterate	Pastoralist	Married	6
04	35	Female	Primary	Pastoralist	Married	3
05	39	Female	Illiterate	Pastoralist	Married	6
06	52	Female	Illiterate	Pastoralist	Married	9

**Table 4 ijerph-18-12518-t004:** Codes, themes, and categories that emerged in the analysis, Afar region, Ethiopia.

S.No	Codes (Problems)	Themes	Subcategories	Major Categories
1.	-long waiting time -discrimination-shouting at client -non-consensual care -lack of greeting -no clarity of information -minimal social support -poor hygiene -water shortage -lack of privacy -rude behaviour -non-dignified care	-low accountability -disrespectful care -non-dignified care -absence of autonomous care -breach of confidentiality -communication problems -absence of social needs -absence of informed decision -less accountability -quality of basic amenities -negligence	-dignified care -autonomous care -proper communication -prompt attention -social support provision -quality of basic amenities -confidential care	responsiveness of care
2.	-frequent vaginal exams -lack of knowledge -home delivery -male midwife attendance -wrong religious faith -low community ownership -birth as natural process	-traditional/cultural beliefs -low involvement of clan/religious leaders -perceptions of regret -shame attributed to culture -sex preference -cultural food serving “ounor”	-culturally acceptable care -user friendly care -user-centred approach -TBAs -birth accompanying -ownership	cultural acceptability of care
3.	-care by male midwife -squatting position preference -trust in TBAs -negligence -absence of waiting areas	-sex of service provider -squatting position preference -care preference -social ceremony -community ownership -feeding service -maternity waiting homes	-TBA–SBA partnership -TBA referred care -TBA involvement in companionship -support for TBAs -maternity waiting homes	preferred care modality

## Data Availability

The datasets used in this article are not readily available because the audio recordings contain personal information and are often difficult to anonymize.

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
