# Peer review of "Lived Experiences and Perceptions of Childbirth among Pastoralist Women in North-Eastern Ethiopia: A Multimethod Qualitative Analysis to the WHO Health Systems Responsiveness Framework"

_ijerph, 2021, doi:10.3390/ijerph182312518_

Round 1

Reviewer 1 Report

Please see the comments in the attached document.

Author Response

Please see the attachment below.

Thanks you!

Reviewer 2 Report

In this manuscript, the authors studied the lived experiences and perceptions of childbirth among pastoralist women in north-east Ethiopia. Overall, this study is interesting and may be of potential use.

Major concerns:

-The authors focused on the perception of women who delivered at health facilities. How about the health givers?  How do they respond to the similar events.

-To make the analysis objective, the authors should also include the beneficial aspects/necessity of health facilities.

Minor concern:

-Is there any constructive suggestion to improve the experiences for women in labor at the health facilities?

Author Response

Please see the attachment below.

Thank you!

Reviewer 3 Report

This is a well-written manuscript about a clinically relevant topic. The quotes are impressive. However, I have one major problem and some other comments.

  1. It seems that many of the problems of pastoral women (if that is the correct term) with birth care signalized in this study are already mentioned in the Introduction as found in earlier studies. So, I wondered why the authors started this new study. Can they explain this in the Introduction?

  1. Please explain why the study was carried out in The Afar region.

  1. R. 137-139, " Qualitative methods are appropriate to capture the voices of specific populations for measurable as well as unmeasurable outcomes". This is a strange statement. How can one do a study using unmeasurable outcomes? Also somewhat strange are the statements: "… to capture phenomena by enabling observed and unobserved experiences" and ".. the best method for describing the perceptions by capturing local needs" (R. 103-105). What do these sentences mean? Please check the text for hard-to-understand jargon

  1. R. 161-164, "Data collection was done through key informant interviews, in-depth interviews, focus group discussions, and focused observation". It is easier to absorb all information directly as a reader by saying: "Data collection was done through four key informant interviews, six in-depth interviews, six focus group discussions, and focused observations". Mention directly after this sentence: "The key informants were ….., who were chosen because of ….. The in-depth interviews were done with pastoral women, who were chosen on the basis of …. The focus groups also consisted of pastoral women – 26 in total - who were chosen on the basis of …. The focused observations were done at … locations (which types?). Each observation took …. minutes/hours.

         In general: The text could be better structured.

  1. Please, use less abbreviations and use them at the right places. For instance:

- In the Abstract, the terms IPA and SBA are unnecessary. SBA is not even explained.

- R. 51: Why is the abbr. SSA necessary?

- R. 56-57: The use of SDGs is neither necessary. You can say: "…, especially goal 3 and 10".

- R. 65: The abbr. SBA is explained for the second time.

- R. 84: The abbr. HSR is not necessary.

- R. 136: RIF

- R. 142: IPA

and so on.

  1. The Abstract can be shortened, and the same goes for the introduction. There is, for instance a repetition in the paragraph starting at R. 93.
  2. R. 167: Start a new paragraph at "All interviews …". Check the manuscript for other possibly necessary paragraph format changes.

  1. R. 171: I could not find the Supplementary file.

  1. R. 207-209, "… we validated …. the descriptive phenomenon by comparing it to the standard HSR domains of WHO": Why would your (interpreted) findings be more valid if they are comparable to the HSR domains of WHO? Should it be "described phenomenon"? R. 229-230, "In case of a discrepancy [with the HSR domains of WHO], the research team reviewed the file and re-analyzed the disagreement". Why should there be agreement?

  1. R. 224-225, "Respective multidisciplinary disciplines": What are "multidisciplinary disciplines"? Do you mean "various disciplines of the members of the research team"? What do you mean by "Respective"? What were the disciplines?

  1. R. 240-241, "The KII and IDI were conducted individually and in private": This is so hard to read. Why not simply "The interviews". You could add "The focus groups were held at ….".

  1. Conclusions and recommendations: This should be shortened to about half of its length

The findings of this study deserve a wider audience. I hope the authors will make an effort to have shorter versions of this article published in medical and nursing journals in Ethiopia and will give presentations in hospitals and government institutions.

Author Response

Please see the attachment below

Thnak you!
